# Awareness and its determinant factors towards breast examination to detect breast cancer among reproductive age women in Kenya: Multi level analysis of the recent demographic and health survey data

**Mulugeta Wassie**[1]*, **Alebachew Ferede Zegeye**[2], **Agazhe Aemro**[2], **Bewuketu Terefe**[3], **Gebreeyesus Abera Zeleke**[4], **Belayneh Shetie Workneh**[5], **Berhan Tekeba**[6], **Mohamed Seid Ali**[6], **Enyew Getaneh Mekonen**[4], **Tadesse Tarik Tamir**[6]

1 School of Nursing, College of Medicine and Health Sciences University of Gondar, Gondar, Ethiopia, 2 Department of Medical Nursing, School of Nursing, College of Medicine and Health Sciences University of Gondar, Gondar, Ethiopia, 3 Department of Community Health Nursing, School of Nursing, College of Medicine and Health Sciences University of Gondar, Gondar, Ethiopia, 4 Department of Surgical Nursing, School of Nursing, College of Medicine and Health Sciences University of Gondar, Gondar, Ethiopia, 5 Department of Emergency and Critical Care Nursing, School of Nursing, College of Medicine and Health Sciences University of Gondar, Gondar, Ethiopia, 6 Department of Pediatrics and Child Health Nursing, School of Nursing, College of Medicine and Health Sciences, University of Gondar, Gondar Ethiopia

* mulugeta2113@gmail.com

## Abstract

### Introduction

Worldwide, breast cancer accounts for about 2.3 million new cases and 685,000 deaths, making it the second most common cancer in women. It can be detected through breast examination like mammograms, self-breast examination and clinical breast examinations in the early stage potentially before it spreads to other organs. The current study aimed to determine the awareness of breast examination to detect breast cancer and its determinant factors.

### Methods

A total of 16,474 women of reproductive age were included. The 2022 Kenyan demographic and health survey dataset were used. The data were analyzed using STATA version14.0. Multilevel mixed-effects logistic regression was employed to identify potential factors of the outcome variable. Variables with p-values <0.05 were considered statistically significant.

### Result

Nearly 44.4% of women were aware of breast examinations to detect breast cancer. Older age [AOR = 2.517, 95%CI: 2.11–3.00], higher educational level [AOR = 5.755, 95%CI: 4.631–7.153], being an active worker [AOR = 1.343, 95%CI:1.232–1.465], being the wealthiest [AOR = 1.526, 95%:1.336–1.744], use of the internet [AOR = 1.451, 95%CI:1.323–

**Data Availability Statement:** All relevant data are within the paper and its Supporting Information files.

**Funding:** The author(s) received no specific funding for this work.

**Competing interests:** The authors have declared that no competing interests exist.

1.592], exposure to different media [AOR = 1.350, 95%CI:1.187–1.536] and health facility contact within 12 months [AOR = 1.496, 95%CI:1.385–1.615] were positively associated at the individual level, while low community poverty [AOR = 0.777, 95%:0.676–0.892] and high literacy level [AOR = 1.412, 95%CI:1.190–1.677] were significant factors at the community level.

## Conclusion and recommendation

Less than half of the women were aware of breast examination to detect breast cancer. Older age, high level education, being active worker, exposure for media, using internet, being rich were favorable covariates at the individual level while high literacy and low poverty were found to be enabling factors at the community level. This study recommends that educating women, engaging them in different occupational activities, decreasing poverty, improving media and internet accessibility can bring positive change of women's awareness towards breast examination.

## Introduction

Breast cancer is the second most common cancer type among women accounting 2.3 million new cases and 685,000 deaths in the globe [1]. The Sub-Saharan Africa (SSA) region has the highest incidence rate which shares more than 17.3 per hundred thousand per year. In this region, Southern and Western African take the highest portion with the incidence of 38.9 and 38.6per hundred thousand per year, respectively [2,3].

In Kenya, breast cancer is the most prevalent cancer type, accounting for 16% incidence and 11% cancer-related mortalities. It is ranked as the second most common cause of cancer-related deaths in the country [4].

Being in older age, personal history of breast cancer or benign breast disease, inherited risk of breast cancer, dense breast tissue, greater exposure to estrogen, taking hormonal therapy for symptoms of menopause, radiation therapy to the breast or chest, obesity and drinking alcohol are some of the contributing factors for breast cancer [5,6].

Although the sub-Saharan Africa has the highest rates of breast cancer mortality, the region's breast cancer screening practice is quite low and varies considerably between its nations [7]. Evidences show that attaining higher educational level, being in older age, good health insurance coverage, and highest socioeconomic levels are enabling factors for breast cancer examination in the region [7–9].

In this region, women's health-seeking behavior in regard to breast cancer is also influenced by socio-cultural aspects such as beliefs, traditions, and fear. In addition, poor knowledge for early detection and negative perception of the disease are also another obstacle in the region [9,10].

Unless the prevention strategies are appropriately implemented, the future burden of breast cancer is predicted to increase over 3 million new cases and 1 million deaths in 2040 [2].

Breast cancer can be detected through mammograms, self-examination, and clinical examination at early stage, potentially even before it spreads to different organs [1,11]. To make it true, global efforts are needed to counteract its growing burden, especially in developing countries where incidence is rising rapidly and mortality remains high [12,13].

Precision prevention strategies that can also identify women at high risk and target them for specific therapies, such as risk-minimizing drugs, and population-based strategies that can lower the exposure to modifiable risk factors are needed to reduce the incidence of it [14,15].

Providing pertinent education and socially acceptable awareness creation programs will help to tackle this devastating disease [16]. Low level of breast cancer screening practice is mostly attributed to low public awareness in addition to variety of social, psychological, and geographical barriers [17]. Another evidences also revealed that lack of awareness towards early detection, poor perception of the disease, and socio-cultural factors such as beliefs, traditions, and fear are some of the reasons impacting women's health-seeking behavior in Africa [18,19].

The most important aspect of treating breast cancer is early detection. It can improve patient survival, lower mortality, and decrease national and individual-based health care expenditures [20]. Early detection can be done through breast self-examination(BSE),clinical breast examination(CBE), and breast x-ray (mammography) in which all depends on women's awareness of breast examinations to detect breast cancer [21,22].

The Kenyan national cancer screening initiative identified breast cancer as one of the cancers planned for population-based screening [23]. Since breast cancer screening is the top prioritized problem in the country, the Kenyan national cancer control program tested the viability of using mammography in the county's referral facilities to support a nationwide breast cancer screening program through a breast cancer awareness and screening pilot [24].

Therefore, the current study aimed to determine the current awareness and determinant factors of reproductive-age women towards breast examination to detect breast cancer in Kenya using the most recent demographic and health survey data. This study will be used to estimate the awareness of breast screening in sub-Saharan Africa since there is no national survey dataset in this region except the current setting.

## Methods and materials

### Study design, area, and period

The 2022 Kenyan demographic and health survey (DHS) dataset was used to carry out the current study. It is the country's seventh national population-based cross-sectional survey. Kenya is an African country located in the eastern part of the continent, with latitude of 0.0236°S and a longitude of 37.9062°E. It is bounded by five other African countries, including Ethiopia, Somalia, Uganda, Tanzania, and South Sudan. Currently, the population of Kenya accounts for about 51,629,122. Home to 0.68% of the world's total population, it is the twenty-seventh most populated country in the world [25,26]. The Kenyan DHS is a national survey that collects information on a variety of maternal and child health-related topics, such as HIV/AIDS, tuberculosis, malaria, nutrition, mortality, and domestic violence. In addition, the survey also included information on breast and cervical cancer screening services, as well as women's awareness [27].

DHS uses a two-stage stratified cluster design that includes enumeration areas as the first stage and generates a sample of households from each enumeration area as the second stage [28]. A total sample of 16,474 women aged 15–49 years was included in this study.

### Study variables

**Dependent variables.** In this study, the dependent variable was awareness of breast examination to detect breast cancer. It was coded as "Yes = 1" if the study participants were aware of breast examination to detect breast cancer and "No = 0" if the study participants weren't aware of breast examination to detect breast cancer.

**Independent variables.** Since DHS data are hierarchical in nature, explanatory variables from two sources (at the individual and community levels) were considered. Individual-level variables were age (15–19, 20–24, 25–29, 30–34, 35–39, 40–44, 45–49), educational level (no

formal education, primary, secondary, and higher), occupation (worker, non-worker), marital status (married, unmarried), use of the internet (yes, no), wealth index (poor, middle, rich), media exposure (yes, no), distance from a health facility (not big problem, big problem), pregnancy (yes, no), current health status (good, moderate, bad), information about non-communicable disease (yes, no), visited a health facility in the last 12 months (yes, no). At the community level, place of residence (urban or rural), community literacy (low or high), level of poverty (low or high), and media exposure (low or high) were used to analyze the outcome variable in relation to the explanatory variables.

## Data processing and statistical analysis

The data extracted from DHS datasets was recoded and analyzed using STATA version 14.0. Since DHS data are hierarchal, the best-fit model was found to be multilevel mixed-effects logistic regression to determine the potential factors associated with awareness of breast examination. Multilevel mixed effect logistic regression follows four models: the null model (outcome variable only), mode I (individual level variables with the outcome variable), model II (community level variables with the outcome variable), and model III (both individual and community level variables with the outcome variable). The null model was used to check the variability of the outcome variable across the clusters. In the final model (Model III), the association of both individual and community-level variables was fitted simultaneously. Variables with p-values <0.05 were considered statistically significant at a 95% confidence interval with their corresponding odds ratio. Accordingly, age, educational level, working status, wealth index, use of the internet, media exposure, and visiting health facilities within 12 months were significantly associated with the outcome variable at the individual level, while poverty and literacy level were significantly associated at the community level.

## Random effects and model fitness

To quantify the difference across clusters, the proportionate changes in variance (PCV) and intra-class correlation coefficient (ICC) were calculated. Taking clusters as a random variable, the ICC reflects the variation of breast exam awareness between clusters and it is calculated as; $ICC = \frac{VC}{VC+3.29} \times 100\%$. The PCV shows variation in the outcome variable explained through independent covariates and it is obtained as; $PCV = \frac{Vnull-VC}{Vnull} \times 00\%$; where $V_{null}$ = variance of the null model and VC is cluster level variance.

The fixed effects were used to estimate the association between the likelihood of the outcome variable and individual and community-level covariates. Because of the nested nature of the model, deviation = -2 (log likelihood) was used to compare models, and the model with the lowest deviance was selected as the best-fit model [28].

**Ethical approval and consent to participate.** After the consent letter was submitted to the DHS Programmers/ICF to download the dataset for this investigation, the International Review Board of Demographic and Health Surveys (DHS) programme data archivists' waived informed consent. It is not an experimental study since the datasets came from a publicly accessible source. All the methods were conducted according to the Helsinki Declarations. More details regarding DHS data and ethical standards are available online at (http://www.dhsprogram.com).

## Results

### Socio demographic characteristics of study participants

A total of 16,474 women aged 15–49 years were included in this study. About 18% (2983) of the participants were in the age group of 20–24, and 8% were in the age group of 45–49 years.

Nearly 39% (6412) attended secondary school, more than half (59.28%) had some occupation, and about half (51.35%) were not married. One-third of the participants were poor in the wealth index; nearly only one-tenth (11.61%) had no media exposure, but less than half (46.91%) enjoyed the internet. About a quarter of the participants (23.87%) had a big problem accessing health facilities; 3% described their health status as bad; and almost all (99.64%) had no information about non-communicable diseases. Nearly 60% were living on the rural side of the country. At the community level, about 41% were in high poverty, 81% had high media exposure, and 11% were in low literacy (**Table 1**).

## Awareness of women towards breast examination to detect breast cancer

About 44.37% (95%CI: 43.61–45.13) of women were aware of breast examinations to detect breast cancer in the current study.

## Determinants of awareness towards breast examination to detect breast cancer

A multilevel mixed-effect logistic regression model was used to identify the potential factors that could affect the outcome variable. In the final model, the association of both individual and community-level variables was fitted simultaneously with the outcome variable. In this model, variables with a p-value less than 0.05 with their corresponding AOR were declared statistically significant at a 95% confidence interval.

In the final fitted model, age, educational level, working status, wealth index, use of the internet, media exposure, and visiting health facilities within 12 months were significantly associated at the individual level. At the community level, community poverty and literacy were significantly associated with awareness of breast examinations to detect breast cancer.

Accordingly, the odds of awareness of women in the age group of 45–49 years were about 2.52 [AOR = 2.517, 95% CI: 2.11–3.00] more than those in the age groups of 15–19 years. Women with higher education levels were about 5.76 times more aware of breast examinations to detect breast cancer than those with no education [AOR = 5.755, 95% CI: 4.631–7.153]. The probability of awareness towards breast examination among active workers was about 1.34 [AOR = 1.343, 95%CI: 1.232–1.465] times greater than their counterparts.

Rich women were about 1.53 times more aware towards breast examination to detect breast cancer than poor women [AOR = 1.526, 95%:1.336–1.744]. Internet users were about 1.45 [AOR = 1.451, 95%CI: 1.323–1.592] times more aware of breast examination to detect breast cancer than their counterparts. Women who visited health facility within the last 12 months were about 1.50 times more aware than those who didn't [AOR = 1.496, 95%CI: 1.385–1.615].

At the community level, women with low poverty level were about 22.3% more aware of breast examination to detect breast cancer than those with high poverty level [AOR = 0.777, 95%: 0.676–0.892] and women with high literacy level were about 1.41 times more ware of breast examination to detect breast cancer than those with low literacy level [AOR = 1.412, 95%CI: 1.190–1.677] (**Table 2**).

## Model comparison and random effect

Findings from the null model showed that there are significant variations in awareness of breast examination to detect breast cancer between the clusters, with a variance of 0.9572198 and a P value of 0.000. The variance within clusters contributed 91.22% of the variation of the outcome variable, while the variance across clusters was responsible for 8.78% of the variation.

The interclass correlation value for Model I revealed that 19.63% of the variation accounts for community differences. Cluster variations were the basis for 11.79% of the differences in

**Table 1. Socio- demographic characteristics of study participants.**

| Covariates | Weighted frequency | Weighted percentage |
|---|---|---|
| Age | | |
| 15–19 | 3119 | 18.9% |
| 20–24 | 2983 | 18.1% |
| 25–29 | 2852 | 17.3% |
| 30–34 | 2336 | 14.2% |
| 35–39 | 2261 | 13.7% |
| 40–44 | 1589 | 9.7% |
| 45–49 | 1334 | 8.1% |
| Educational level | | |
| No education | 918 | 5.6% |
| Primary | 6075 | 36.9% |
| Secondary | 6412 | 38.9% |
| Higher | 3069 | 18.6% |
| Occupation | | |
| Non worker | 6709 | 40.7% |
| Worker | 9766 | 59.3% |
| Marital status | | |
| Married | 8015 | 48.7% |
| Single | 8459 | 51.4% |
| Wealth index | | |
| Poor | 5549 | 33.7% |
| Middle | 3063 | 18.6% |
| Rich | 7862 | 47.7% |
| Media exposure | | |
| Yes | 14561 | 88.4% |
| No | 1913 | 11.6% |
| Internet use | | |
| Yes | 7729 | 46.9% |
| No | 8745 | 53.1% |
| Distance from health facility | | |
| Not big problem | 12542 | 76.1% |
| Big problem | 3932 | 23.9% |
| Pregnancy | | |
| Yes | 919 | 5.6% |
| No | 15555 | 94.4% |
| Health status | | |
| Good | 12833 | 77.9% |
| Moderate | 3216 | 19.5% |
| Bad | 425 | 2.6% |
| NCD information | | |
| Yes | 60 | 0.4% |
| No | 16415 | 99.6% |
| HF visited in 12 month | | |
| Yes | 8893 | 54.0% |
| No | 7581 | 46.0% |
| Residency | | |
| Urban | 6673 | 40.5% |

(*Continued*)

**Table 1.** (Continued)

| Covariates | Weighted frequency | Weighted percentage |
|---|---|---|
| Rural | 9801 | 59.5% |
| Community poverty level | | |
| Low | 9694 | 58.8% |
| High | 6780 | 41.2% |
| Community media exposure | | |
| Low | 3077 | 18.7% |
| High | 13397 | 81.3% |
| Community literacy | | |
| Low | 1780 | 10.8% |
| High | 14694 | 89.2% |

*NCD = non communicable disease.*

awareness of breast examination from the ICC value from Model II. In the final model, 60.58% awareness of breast examination was attributed to individual and community-level variables (**Table 3**).

## Discussion

The current study aimed to determine the awareness and determinant of breast examination to detect breast cancer in Kenya. This study revealed that only about 44.4% of women were aware of breast examination to detect breast cancer. This result is an alarming sign for policy-makers and other stakeholders to implement socio-culturally acceptable ways of awareness-creation modalities to improve breast examinations to detect this disease. When comparing to other studies, this finding is less than the studies conducted in Namibia with 60% [29], Saudi Arabia, 57.9% [30] and Nepal, 82% [31] but higher than Tricky, Tamil Nadu 26% [32] and Iraq,(30.3%) [33].

The result differences might be due to the sample size differences as the current study is a national-based survey compared to other studies. In addition, the sociocultural differences of the given countries might be the contributing factors of the disparities.

The current study shows that age, educational level, working status, wealth index, use of the internet, media exposure, and visiting health facilities within 12 months were significantly associated at the individual level. At the community level, community poverty and literacy level were significantly associated with awareness of breast examinations.

In this study, as the women get older and older, their awareness of breast examination also increased simultaneously. The current finding is in line with the study conducted in Lesotho [34], but contradicts with the study conducted in China [35]. The similarity of these findings could be explained as the older women could have more contact with health facilities for other reasons such as antenatal care and family planning. The probability of getting information about breast examination may be increased if there is frequent contact of health facilities [36,37]. This might be also true in the current finding. To narrow this gap, it is strongly advisable to strengthen youth-friendly clinics led by skilled health care providers who can provide health education related to breast cancer and its prevention mechanisms for the young ladies.

More educated women were found to have good awareness towards breast examination to detect breast cancer in this study. The current study is in line with other studies conducted in different settings [38–41]. The possible explanation could be as women get more educated, their health awareness level can be increased too. This result shows that improving women's

**Table 2. Multilevel logistic regression analysis of individual and community level factors.**

| Variables | Null model | Model I [AOR(95% CI)] | Model II [AOR(95% CI)] | Model III [AOR(95% CI)] |
|---|---|---|---|---|
| Age | | | | |
| 15–19 | | 1 | | 1 |
| 20–24 | | 1.15 (1.01–1.31) | | **1.14 (1.00–1.30)** |
| 25–29 | | 1.39 (1.20–1.61) | | **1.36 (1.18–1.58)** |
| 30–34 | | 1.82 (1.56–2.12) | | **1.76 (1.51–2.05)** |
| 35–39 | | 2.38 (2.04–2.78) | | **2.29 (1.96–2.68)** |
| 40–44 | | 2.32 (1.96–2.74) | | **2.24 (1.89–2.64)** |
| 45–49 | | 2.62 (2.19–3.12) | | **2.52 (2.11–3.00)** |
| Educational level | | | | |
| No education | | 1 | | 1 |
| Primary | | 2.27 (1.92–2.69) | | **1.87 (1.55–2.24)** |
| Secondary | | 3.22 (2.69–3.86) | | **2.60 (2.14–3.16)** |
| Higher | | 7.06 (5.76–8.67) | | **5.755(4.63–7.15)** |
| Occupation | | | | |
| Non worker | | 1 | | 1 |
| Worker | | 1.38 (1.26–1.50) | | **1.34 (1.23–1.46)** |
| Marital status | | | | |
| Married | | 1.02(0.94–1.12) | | 1.05 (0.96–1.14) |
| Non married | | 1 | | 1 |
| Wealth index | | | | |
| Poor | | 1 | | |
| Middle | | 1.37 (1.23–1.53) | | **1.26 (1.12–1.41)** |
| Rich | | 1.78(1.59–1.98) | | **1.52 (1.33–1.74)** |
| Media exposure | | | | |
| Yes | | 1.43 (1.26–1.62) | | **1.35 (1.19–1.54)** |
| No | | 1 | | 1 |
| Internet use | | | | |
| Yes | | 1.45 (1.32–1.59) | | **1.45(1.32–1.59)** |
| No | | 1 | | 1 |
| Distance from health facility | | | | |
| Not big problem | | 1.05 (0.96–1.15) | | 1.03 (0.94–1.13) |
| Big problem | | 1 | | 1 |
| Currently pregnant | | | | |
| Yes | | 1.02 (0.86–1.19) | | 1.02 (0.87–1.20) |
| No | | 1 | | 1 |
| Health status | | | | |
| Good | | 1 | | 1 |
| Moderate | | 1.03 (0.94–1.13) | | 1.03(0.93–1.13) |
| Bad | | 0.97(0.77–1.23) | | 0.97(0.77–1.22) |
| NCD information | | | | |
| Yes | | 1.14 (0.68–1.88) | | 1.22 (0.73–2.02) |
| No | | 1 | | |
| Visited HF in 12 month | | | | |
| Yes | | 1.49 (1.38–1.62) | | **1.49(1.38–1.61)** |
| No | | 1 | | 1 |
| **Community level** | | | | |

*(Continued)*

**Table 2.** (Continued)

| Variables | Null model | Model I [AOR(95% CI)] | Model II [AOR(95% CI)] | Model III [AOR(95% CI)] |
|---|---|---|---|---|
| Residency | | | | |
| Urban | | | 1.21 (1.07–1.37) | 0.97 (0.85–1.11) |
| Rural | | | 1 | 1 |
| Community poverty level | | | | |
| Low | | | 1 | 1 |
| High | | | 0.50 (0.44–0.57) | **0.77 (0.67–0.89)** |
| Community media exposure | | | | |
| High | | | 1.46 (1.27–1.68) | 1.05 (0.91–1.21) |
| Low | | | 1 | 1 |
| Community literacy | | | | |
| High | | | 2.56 (2.18–3.01) | **1.41 (1.19–1.67)** |
| Low | | | | 1 |

*Bold fonts in the above table indicate significantly associated variables.

educational level would have a positive impact on breast exam awareness to detect breast cancer.

Women who were active workers during the data collection period were more aware of breast examination to detect breast cancer as compared to non-active workers. These could be due to the fact that employed women might be recruited from more educated ones. This argument is just like more educated women are more aware of breast examinations and employed women might be by default more educated. This finding implies that engaging and incentivizing women in different sectors like the private and governmental sectors can raise the awareness of breast examination.

Women who had media exposure and used the internet were more aware of breast examination than their counterparts. This finding is supported by different studies [42–45]. This might be due to the fact that women who are exposed for different media and internet could explore different websites and podcasts that work with cancer prevention and treatment options and resulted with good awareness. This result implies that increasing the accessibility of health journals like radio, television, and magazines is expected from the ministry policy-makers of the country to increase the awareness level of the women towards breast examination.

Those women who visited health facilities within 12 months were more aware of breast examination in the current study. This could mean that women with frequent visits to health

**Table 3. Model comparison and random effect.**

| Parameter | Null model | Model I | Model II | Model III |
|---|---|---|---|---|
| Variance | 0.9572198 | 0.3805995 | 0.4400797 | 0.3773161 |
| ICC | 19.63% | 10.37% | 11.79% | 8.78% |
| PCV | Reference | 60.24% | 54.03% | 60.58% |
| **Model fitness** | | | | |
| LL | -10620.653 | -9482.4855 | -10244.563 | -9459.7241 |
| Deviance | 21241.306 | 18964.971 | 20489.126 | 18919.4482 |

ICC: Intera cluster correlation, LL: Logliklihood, PCV: Proportional change in variance.

facilities would receive information regarding different breast examination techniques to detect breast cancer. However, we couldn't find any other data to compare and contrast this finding. The current finding suggests that community based health education regarding to breast cancer and its prevention mechanisms is highly advisable.

At the community level, women with low poverty levels were more aware of breast examination than those with high poverty levels in the current study. This finding could be justified as women with low poverty levels could access different mass media (radio, television, and magazines) and resulted in an increase of their level of awareness of breast examination. As explained in the previous sections, the women who exposed to internet and different media would get different information regarding breast cancer and its prevention mechanisms. This result implies that increasing women's economic level can dramatically increase the awareness level of the women towards breast examination to detect breast cancer.

Similarly, women with a high level of literacy were more aware of breast examinations to detect breast cancer than their counterparts. The reasons and policy implications are justified in the previous paragraphs that describe individual-level educational status.

## Conclusion

Less than half of the women were aware of breast examination to detect breast cancer in the current study. This study found that the current problem is public health important. Older age, high-level education, being a worker, media exposure, using the internet, visiting a health facility within the last twelve months, and being rich were found to be enabling conditions for being aware of breast examination at the individual level. At the community level, high community literacy and low poverty levels were also enabling factors. This study recommends that educating women at the individual and community level, engaging in different income generating activities, decreasing individual and community poverty, creating awareness using different media and improving internet service in the country will bring positive change.

## Author Contributions

**Conceptualization:** Mulugeta Wassie, Alebachew Ferede Zegeye, Belayneh Shetie Workneh, Berhan Tekeba, Tadesse Tarik Tamir.

**Formal analysis:** Mulugeta Wassie, Alebachew Ferede Zegeye, Bewuketu Terefe, Tadesse Tarik Tamir.

**Investigation:** Mulugeta Wassie, Agazhe Aemro, Gebreeyesus Abera Zeleke, Berhan Tekeba.

**Methodology:** Mulugeta Wassie, Alebachew Ferede Zegeye, Agazhe Aemro, Bewuketu Terefe, Gebreeyesus Abera Zeleke, Mohamed Seid Ali, Enyew Getaneh Mekonen, Tadesse Tarik Tamir.

**Project administration:** Mulugeta Wassie.

**Resources:** Mulugeta Wassie.

**Software:** Mulugeta Wassie, Agazhe Aemro, Belayneh Shetie Workneh, Enyew Getaneh Mekonen, Tadesse Tarik Tamir.

**Supervision:** Mulugeta Wassie.

**Validation:** Mohamed Seid Ali.

**Visualization:** Mulugeta Wassie, Alebachew Ferede Zegeye, Bewuketu Terefe, Gebreeyesus Abera Zeleke, Belayneh Shetie Workneh, Berhan Tekeba, Mohamed Seid Ali, Enyew Getaneh Mekonen, Tadesse Tarik Tamir.

**Writing – original draft:** Mulugeta Wassie, Alebachew Ferede Zegeye, Agazhe Aemro, Bewuketu Terefe, Gebreeyesus Abera Zeleke, Belayneh Shetie Workneh, Berhan Tekeba, Mohamed Seid Ali, Enyew Getaneh Mekonen, Tadesse Tarik Tamir.

**Writing – review & editing:** Mulugeta Wassie, Alebachew Ferede Zegeye, Agazhe Aemro, Bewuketu Terefe, Gebreeyesus Abera Zeleke, Belayneh Shetie Workneh, Berhan Tekeba, Mohamed Seid Ali, Enyew Getaneh Mekonen, Tadesse Tarik Tamir.

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
