## [Decision Letter · Decision Letter 0]

3 Jul 2024

PONE-D-24-24195Awareness and its determinant factors towards breast examination to detect breast cancer among reproductive age women in Kenya: Multi level analysis of the recent Demographic and Health Survey dataPLOS ONE

Dear Dr. Wassie,

Thank you for submitting your manuscript to PLOS ONE. After careful consideration, we feel that it has merit but does not fully meet PLOS ONE’s publication criteria as it currently stands. Therefore, we invite you to submit a revised version of the manuscript that addresses the points raised during the review process.

 Please read the reviewers' and editor's comments carefully, and address all comments made by them.

Please submit your revised manuscript by Aug 17 2024 11:59PM. If you will need more time than this to complete your revisions, please reply to this message or contact the journal office at plosone@plos.org. Please include the following items when submitting your revised manuscript:A rebuttal letter that responds to each point raised by the academic editor and reviewer(s). You should upload this letter as a separate file labeled 'Response to Reviewers'.A marked-up copy of your manuscript that highlights changes made to the original version. You should upload this as a separate file labeled 'Revised Manuscript with Track Changes'.An unmarked version of your revised paper without tracked changes. You should upload this as a separate file labeled 'Manuscript'.

We look forward to receiving your revised manuscript.

Kind regards,

Essa Tawfiq

Academic Editor

PLOS ONE

Journal Requirements:

https://www.cell.com/heliyon/fulltext/S2405-8440(20)30598-3?_returnURL=https%3A%2F%2Flinkinghub.elsevier.com%2Fretrieve%2Fpii%2FS2405844020305983%3Fshowall%3Dtrue

In your revision ensure you cite all your sources (including your own works), and quote or rephrase any duplicated text outside the methods section. Further consideration is dependent on these concerns being addressed.

3. We note that your Data Availability Statement is currently as follows: 

"All relevant data are within the manuscript and its Supporting Information files."

**Additional Editor Comments:**

The study examined women's awareness of breast cancer examination and associated factors in Kenya. This is very important that women have knowledge of the availability of such health services to seek care on breast cancer examination which can assist in early detection and timely treatment and surgical intervention of breast cancer. The findings from this study may have policy implications to improve care seeking behavior for breast cancer examination among women through designing and implementing evidence-based healthcare interventions at community level. The authors need to address all comments made the reviewers in order to improve the manuscript.

Reviewers' comments:

Reviewer's Responses to Questions

**Comments to the Author**

1. Is the manuscript technically sound, and do the data support the conclusions?

Reviewer #1: Yes

Reviewer #2: Yes

2. Has the statistical analysis been performed appropriately and rigorously? 

Reviewer #1: Yes

Reviewer #2: Yes

3. Have the authors made all data underlying the findings in their manuscript fully available?

Reviewer #1: Yes

Reviewer #2: Yes

4. Is the manuscript presented in an intelligible fashion and written in standard English?

Reviewer #1: Yes

Reviewer #2: No

5. Review Comments to the Author

Reviewer #1: Review comments

Thank you for the opportunity to review this paper. The title of the paper is relevant and the findings has the potential to the health of women in Kenya and Africa at large. Kindly find my comments below

General comments

1. Authors need to reorganize the introduction and discussion sections into meaningful paragraphs to direct the readers

2. I suggest the authors make reference to studies published in Africa more in the discussion section to support the study context.

3. Authors should improve the discussion section by stating their findings, providing reasons and recommendations but not just stating findings that otherwise support or not support their findings. This will provide further insight to the readers

4. Manuscript editing and formatting is needed to improve reading

Abstract

Introduction:

5. “Breast cancer is the second most frequent malignancy in women with over 2.3 million new cases and 685,000 deaths worldwide” this statement should be rephrased since it is the same as in the introduction in the main text

Introduction

6. Include data on prevalence of breast cancer (BC) in SSA while also citing relevant studies in Africa including such findings in Kenya.

7. Include data on the awareness level of breast examination in SSA including Kenya if any

8. Also include information of the determinants of breast examination in SSA including Kenya

9. You need to clearly articulate the problem under investigation to give the readers an opportunity to appreciate why you are conducting this study or why this study is relevant.

Methods

10. Include detail information about the setting of the study

11. Include detail information about the source of data. You may look at a similar study published by (Afaya et al., 2023, which used demographic and health survey data from Lesotho) for guidance.

Design

12. Under design, include that this is a national population-based cross-sectional survey

13. Under study design, the statement “aware of exam breast for breast cancer (s1102c)” is not clear. If possible, state this clearly.

Variables

Dependent

14. “The variable aware of exam breast for breast cancer (s1102c)” from the maternal record (IR) dataset was chosen and recoded to create the outcome variable”.

I suggest you rather include this statement in this section instead of the study design. I suggest you also state the actual question that was asked to the participants about their awareness of breast examination.

Results

Demographics

15. The statement “nearly only one tenth (11.61%) hadn’t media exposure”. Is it “had no media” exposure? Rephrase if necessary

Discussions

16. The statement “This could be workers would be privates and government employed recruited from more literate women” is not clear. Kindly check and rephrase for clarity

Reviewer #2: Thank you for giving me the opportunity to review this important topic. Here are my comments.

Strengths:

1. Relevant and important topic

2. Large and updated data set

Weaknesses: The authors should address the following comments before acceptance.

Abstract:

1- Results: The authors should add AOR with 95%CI for significant variables.

2- Conclusion: Please be specific and policy relevant.

Introduction:

- The introduction needs to be comprehensively revises. The authors should add:

- The significance of breast examination in breast cancer detections.

- What was the level and determinants in earlier studies conducted in Kenya, If there are no studies the authors should

focus on other LMICs.

- What is the current national policy? Are there any interventions for improving awareness in Kenya now?

- The authors can also mention on the importance of research.

Methods:

-The authors should mention what variables were selected in MLR.

Results:

- Please write the results of MLR analysis in correct format: [AOR, 95%CI:]. The format used is not similar for all variables.

Discussion:

This needs a bit more work. At current form it is a repetition of the results. A good discussion should have four arms: 1) present key finding, 2) compare with other literature, 3) rational and 4) policy implication. I suggest you revise your discussion based on the above criteria. Present each key finding, compare with earlier literature, and provide policy and practical implications for them. Each finding should have a key implication for practice and policy.

Finally, I would highly recommend that you guys have a native English speaker (someone who grew up in an environment with English as their mother tongue) to review the readability of this article. I do think that you guys have decent English, but it comes across as janky and the flow is quite uneven. Being a non native English speaker I understand the struggle you guys are facing. Unfortunately, there is a limit to how much we can do on our own. It is imperative that we get our articles proofread by a native English speaker (preferably someone from America to ensure wider acceptance). Therefore, I strongly recommend improving the language of the the article.

6. PLOS authors have the option to publish the peer review history of their article (what does this mean?). If published, this will include your full peer review and any attached files.

Reviewer #1: No

Reviewer #2: **Yes: **Muhammad Haroon Stanikzai

---

## [Author Response · Author response to Decision Letter 0]

16 Jul 2024

Date: 10/07/2024

Journal: PLoS ONE

Ref: PONE-D-24-24195

Title: Awareness and its determinant factors towards breast examination to detect breast cancer among reproductive age women in Kenya: Multi level analysis of the recent Demographic and Health Survey data. 

Authors: Mulugeta Wassie1*, Alebachew Ferede Zegeye2 ,Agazhe Aemro2, Bewuketu Terefe3, , Gebreeyesus Abera Zeleke4, Belayneh Shetie Workneh5, Berhan Tekeba6, Mohamed Seid Ali6, Enyew Getaneh Mekonen4 ,Tadesse Tarik Tamir6

Subject: Submission of revised manuscript

Thank you so much for allowing us to revise our manuscript entitled “Awareness and its determinant factors towards breast examination to detect breast cancer among reproductive age women in Kenya: Multi level analysis of the recent Demographic and Health Survey data". We are very pleased for the reviewers’ and editor’s comments. We have carefully revised the manuscript and incorporated the comments accordingly. Our responses are given in point-by-point response below.

We hope the revised version is suitable for publication and look forward to hearing from you in due courses.

Sincerely

Mulugeta Wassie

Corresponding Author

Point by point responses to Reviewers’ comments

Editor’s comments 

Author’s response: Thank you so much for your reminding. We have seen the PLOS ONE's manuscript submission style requirements carefully and tried to prepare the manuscript accordingly.

2. We noticed you have some minor occurrence of overlapping text with the following previous publication(s), which needs to be addressed: https://www.cell.com/heliyon/fulltext/S2405-8440(20)30598-3?_returnURL=https%3A%2F%2Flinkinghub.elsevier.com%2Fretrieve%2Fpii%2FS2405844020305983%3Fshowall%3Dtrue. In your revision ensure you cite all your sources (including your own works), and quote or rephrase any duplicated text outside the methods section. Further consideration is dependent on these concerns being addressed.

Author’s response: Thank you very much for your suggestion. We have addressed your comments and suggestions in the revised manuscript.

3. We note that your Data Availability Statement is currently as follows: 

"All relevant data are within the manuscript and its Supporting Information files."

Author’s response: Thank you for your comment. We have shared the dataset used to extract the results in the supporting information section of the submission system.

Additional Editor Comments:

The study examined women's awareness of breast cancer examination and associated factors in Kenya. This is very important that women have knowledge of the availability of such health services to seek care on breast cancer examination which can assist in early detection and timely treatment and surgical intervention of breast cancer. The findings from this study may have policy implications to improve care seeking behavior for breast cancer examination among women through designing and implementing evidence-based healthcare interventions at community level. The authors need to address all comments made the reviewers in order to improve the manuscript.

Author’s Response: We thank you for your understanding of the aim of this research. We tried to address all the comments and suggestions raised by reviewers as follows: 

Review comments

Reviewer #1: 

Thank you for the opportunity to review this paper. The title of the paper is relevant and the findings have the potential to the health of women in Kenya and Africa at large. Kindly find my comments below

General comments

1. Authors need to reorganize the introduction and discussion sections into meaningful paragraphs to direct the readers

Authors’ response: Thank you so much for your suggestion: We have reorganized the introduction and discussion section carefully in the revised manuscript.

2. I suggest the authors make reference to studies published in Africa more in the discussion section to support the study context.

Authors’ response: Thank for your suggestion. We tried to include African literatures in the discussion section although the researches related to the current title in this region are rare.

3. Authors should improve the discussion section by stating their findings, providing reasons and recommendations but not just stating findings that otherwise support or not support their findings. This will provide further insight to the readers

Authors’ response: Thank you for your suggestion. We have addressed it in the revised manuscript.

4. Manuscript editing and formatting is needed to improve reading

Authors’ response: Thank you for your suggestion and comments. We tried to proofread the whole revised manuscript in the scholarly manner.

Abstract

Introduction:

5. “Breast cancer is the second most frequent malignancy in women with over 2.3 million new cases and 685,000 deaths worldwide” this statement should be rephrased since it is the same as in the introduction in the main text

Authors’ response: Thank you so much for your suggestions. It is corrected in the revised manuscript.

Introduction

6. Include data on prevalence of breast cancer (BC) in SSA while also citing relevant studies in Africa including such findings in Kenya.

Authors’ response: Thank you so much for your comment. It has been included in the revised manuscript.

7. Include data on the awareness level of breast examination in SSA including Kenya if any

Authors’ response: Thank you for your comment. Although there are limited literatures, we tried to address them in the revised manuscript.

8. Also include information of the determinants of breast examination in SSA including Kenya

Authors’ response: Dear reviewer, thank you for your comment. The comments have been addressed in the revised manuscript.

9. You need to clearly articulate the problem under investigation to give the readers an opportunity to appreciate why you are conducting this study or why this study is relevant.

Authors’ response: Dear reviewer, thank you for your comment. The authors have tried to include this comment in the revised manuscript.

Methods

10. Include detail information about the setting of the study

Authors’ response: Thank you for your suggestions. We have included detailed information about the setting in the revised manuscript.

11. Include detail information about the source of data. You may look at a similar study published by (Afaya et al., 2023, which used demographic and health survey data from Lesotho) for guidance.

Authors’ response: Thank you so much for your comments and suggestions. We have included it in the revised manuscript.

Design

12. Under design, include that this is a national population-based cross-sectional survey

Authors’ response: Thank you so much for your comments. It is included in the revised manuscript.

13. Under study design, the statement “aware of exam breast for breast cancer (s1102c)” is not clear. If possible, state this clearly.

Authors’ response: Thank you for your suggestion. It is directly taken as it appears in the DHS dataset. It is rephrased in the revised manuscript.

Variables

Dependent

14. “The variable aware of exam breast for breast cancer (s1102c)” from the maternal record (IR) dataset was chosen and recoded to create the outcome variable”.

I suggest you rather include this statement in this section instead of the study design. I suggest you also state the actual question that was asked to the participants about their awareness of breast examination.

Authors’ response: Thank you for your suggestion. We put it on this section and it is rephrased as “are you aware of breast exam to detect breast cancer”. 

Results

Demographics

15. The statement “nearly only one tenth (11.61%) hadn’t media exposure”. Is it “had no media” exposure? Rephrase if necessary

Authors’ response: Thank you for your suggestion. We have rephrased it in the revised manuscript.

Discussions

16. The statement “This could be workers would be privates and government employed recruited from more literate women” is not clear. Kindly check and rephrase for clarity.

Authors’ response: Thank you for your suggestion. We have seen it and rephrased in the revised manuscript.

Reviewer #2: 

Thank you for giving me the opportunity to review this important topic. Here are my comments.

Strengths:

1. Relevant and important topic

2. Large and updated data set

Weaknesses: The authors should address the following comments before acceptance.

Abstract:

1- Results: The authors should add AOR with 95%CI for significant variables.

Authors’ response: Dear reviewer, thank you so much for your comments.

We have included all these information in the revised manuscript.

2- Conclusion: Please be specific and policy relevant.

Authors’ response: thank you so much for your suggestion. We have tried to revise it in the revised manuscript.

Introduction:

- The introduction needs to be comprehensively revises. The authors should add:

- The significance of breast examination in breast cancer detections.

- What was the level and determinants in earlier studies conducted in Kenya, If there are no studies the authors should focus on other LMICs.

- What is the current national policy? Are there any interventions for improving awareness in Kenya now?

- The authors can also mention on the importance of research.

Authors’ response: thank you so much for your suggestion. We have addressed all the issues raised in the revised manuscript.

Methods:

-The authors should mention what variables were selected in MLR.

Authors’ response: thank you so much for your comments. It is addressed in the “Data processing and statistical analysis” section of the revised manuscript.

Results:

- Please write the results of MLR analysis in correct format: [AOR, 95%CI:]. The format used is not similar for all variables.

Authors’ response: thank you so much for your comments. It is addressed in the revised manuscript.

Discussion:

This needs a bit more work. At current form it is a repetition of the results. A good discussion should have four arms: 1) present key finding, 2) compare with other literature, 3) rational and 4) policy implication. I suggest you revise your discussion based on the above criteria. Present each key finding, compare with earlier literature, and provide policy and practical implications for them. Each finding should have a key implication for practice and policy.

Authors’ response: thank you so much for your comments. We have tried to revise the discussion based on the comments in the revised manuscript.

Finally, I would highly recommend that you guys have a native English speaker (someone who grew up in an environment with English as their mother tongue) to review the readability of this article. I do think that you guys have decent English, but it comes across as janky and the flow is quite uneven. Being a non native English speaker I understand the struggle you guys are facing. Unfortunately, there is a limit to how much we can do on our own. It is imperative that we get our articles proofread by a native English speaker (preferably someone from America to ensure wider acceptance). Therefore, I strongly recommend improving the language of the article.

Authors’ response: Dear reviewer, we really appreciate your deep understanding. Even though still there could be some unseen typos, we tried to improve the language of the revised manuscript.

---

## [Decision Letter · Decision Letter 1]

8 Oct 2024

PONE-D-24-24195R1Awareness and its determinant factors towards breast examination to detect breast cancer among reproductive age women in Kenya: Multi level analysis of the recent Demographic and Health Survey dataPLOS ONE

Dear Dr. Wassie,

Thank you for submitting your manuscript to PLOS ONE. After careful consideration, we feel that it has merit but does not fully meet PLOS ONE’s publication criteria as it currently stands. Therefore, we invite you to submit a revised version of the manuscript that addresses the points raised during the review process.

**ACADEMIC EDITOR: **Please read the reviewers' and editor's comments carefully, and address all comments made by them.

We look forward to receiving your revised manuscript.

Kind regards,

Yitagesu Habtu

Academic Editor

PLOS ONE

Reviewers' comments:

Reviewer's Responses to Questions

**Comments to the Author**

1. If the authors have adequately addressed your comments raised in a previous round of review and you feel that this manuscript is now acceptable for publication, you may indicate that here to bypass the “Comments to the Author” section, enter your conflict of interest statement in the “Confidential to Editor” section, and submit your "Accept" recommendation.

Reviewer #1: (No Response)

Reviewer #2: (No Response)

2. Is the manuscript technically sound, and do the data support the conclusions?

Reviewer #1: Yes

Reviewer #2: No

3. Has the statistical analysis been performed appropriately and rigorously? 

Reviewer #1: Yes

Reviewer #2: Yes

4. Have the authors made all data underlying the findings in their manuscript fully available?

Reviewer #1: Yes

Reviewer #2: No

5. Is the manuscript presented in an intelligible fashion and written in standard English?

Reviewer #1: Yes

Reviewer #2: No

6. Review Comments to the Author

Reviewer #1: I see that the discussion section has a lot of disjointed paragraphs. These paragraphs could be well-aligned or merged to improve the reading and flow of the message. Statements that are talking about the same thing should be put together. Example;

“The result differences might be due to the sample size differences, as the current study is a national-based survey, in addition to the sociocultural differences of the given countries. The current result could be an alarming sign for policymakers and other stakeholders to implement socio-culturally acceptable ways of awareness-creation modalities to improve breast examinations to detect breast cancer in the country”.

This paragraph should be added to the first to make it a complete sentence

Reviewer #2: Dear Authors,

The authors have made several changes that were recommended in the initial review, and the current submission is a much more relevant and interesting article to read. The added discussion of factors that make the women of Kenya uniquely susceptible adds context and relevance to the study. There are still many points that need to be addressed before this manuscript can be accepted for publication.

Major Points:

Introduction: The Introduction is still an issue. As the first part of the article that a potential reader reads, it should be attractive and easy to read. You have to "sell" the article to the reader.

Results: The statistical methods employed should be explained in methods section.

Discussion: Discussion needs more work. Based on the previous comments, the authors should present key findings compare with other studies, possible explanations, and key practical implications.

Minor points:

Abstract:

- Methods: A total of 16,474 women were included. Please revise as: A total of 16,474 women of reproductive age were included.

- Results: Please follow correct spacing. The authors should give a space after comma. The are a lot of spacing and punctuation errors through out the manuscript.

- Conclusion: The authors recommend- Please revise as the study recommends. Please use small letter for Media.

Introduction:

- Please give a space between citation and preceding word. Line: 6: Respectively(2,3). It should be respectively (2,3). Please follow this in whole manuscript. Use your citations as per journal guideline

[2,3].

Methods:

- Study variables: Please remove the variable number and the data set number.

Results:

- weren't married: Were not married

- hadn't any information: had no information

- Please remove dot before . (Table 1).

Table 1. Please use one number after coma (18.93%, make it 18.9%). Please revise all table.

Table 1. What does non married mean?

Table 2. Please two numbers after coma. For example: 1.156 (1.014-1.318). It should be 1.15 (1.01-1.31). Please revise all table.

This manuscript need major editing for language and grammar. At its current form, the language of the article is not appropriate for academic publishing.

In the revised manuscript, please give page and line numbers (using continuous options). The authors also have to revise their data availability statement and how can be data accessed. I believe the data will have a repository link.

Thank you,

7. PLOS authors have the option to publish the peer review history of their article (what does this mean?). If published, this will include your full peer review and any attached files.

Reviewer #1: No

Reviewer #2: **Yes: **Muhammad Haroon Stanikzai

---

## [Author Response · Author response to Decision Letter 1]

15 Oct 2024

Date: 15/10/2024

Journal: PLoS ONE

Ref: PONE-D-24-24195R1

Title: Awareness and its determinant factors towards breast examination to detect breast cancer among reproductive age women in Kenya: Multi level analysis of the recent Demographic and Health Survey data. 

Authors: Mulugeta Wassie1*, Alebachew Ferede Zegeye2 ,Agazhe Aemro2, Bewuketu Terefe3, , Gebreeyesus Abera Zeleke4, Belayneh Shetie Workneh5, Berhan Tekeba6, Mohamed Seid Ali6, Enyew Getaneh Mekonen4 ,Tadesse Tarik Tamir6

Subject: Submission of 2nd revised manuscript

Thank you so much for allowing us to revise our manuscript entitled “Awareness and its determinant factors towards breast examination to detect breast cancer among reproductive age women in Kenya: Multi level analysis of the recent Demographic and Health Survey data". We are very pleased for the reviewers’ and editor’s comments. We have carefully revised the manuscript and incorporated the comments accordingly. Our responses are given in point-by-point response below.

We hope the revised version is suitable for publication and look forward to hearing from you in due courses.

Sincerely

Mulugeta Wassie

Corresponding Author

Point by point responses to Reviewers’ comments

Reviewer #1:

 I see that the discussion section has a lot of disjointed paragraphs. These paragraphs could be well-aligned or merged to improve the reading and flow of the message. Statements that are talking about the same thing should be put together. Example;

“The result differences might be due to the sample size differences, as the current study is a national-based survey, in addition to the sociocultural differences of the given countries. The current result could be an alarming sign for policymakers and other stakeholders to implement socio-culturally acceptable ways of awareness-creation modalities to improve breast examinations to detect breast cancer in the country”.

This paragraph should be added to the first to make it a complete sentence

Authors’ response: Thank you so much for your constructive comments. All the concerns are addressed in the revised manuscript.

Reviewer #2:

 Dear Authors,

The authors have made several changes that were recommended in the initial review, and the current submission is a much more relevant and interesting article to read. The added discussion of factors that make the women of Kenya uniquely susceptible adds context and relevance to the study. There are still many points that need to be addressed before this manuscript can be accepted for publication.

Major Points:

Introduction: The Introduction is still an issue. As the first part of the article that a potential reader reads, it should be attractive and easy to read. You have to "sell" the article to the reader.

Authors’ response: Thank you so much for your unreserved effort and your comments. We tried to incorporate the comments in the main manuscript.

Results: The statistical methods employed should be explained in methods section.

Authors’ response: Thank you so much for your suggestion. The statistical methods used in the results section are explained at the data processing and analysis section (from lines 115 to 124) and Random effects and model fitness section (from lines 129 to 138).

Discussion: Discussion needs more work. Based on the previous comments, the authors should present key findings compare with other studies, possible explanations, and key practical implications?

Authors’ response: Thank you for your constructive comments. We tried to address the comments in the revised manuscript. 

Minor points:

Abstract:

- Methods: A total of 16,474 women were included. Please revise as: A total of 16,474 women of reproductive age were included. 

Authors’ response: Thank you for your comment. It is corrected as per the suggestion given.

- Results: Please follow correct spacing. The authors should give a space after comma. There are a lot of spacing and punctuation errors throughout the manuscript.

Authors’ response: Thank you for your comment. We tried to address the comments in the main revised document. 

- Conclusion: The authors’ recommend- Please revise as the study recommends. Please use small letter for Media.

Authors’ response: Thank you for your recommendation: It is addressed in the revised document.

Introduction:

- Please give a space between citation and preceding word. Line: 6: Respectively(2,3). It should be respectively (2,3). Please follow this in whole manuscript. Use your citations as per journal guideline [2,3].

Authors’ response: Thank you for your suggestion. We tried to incorporate the comments in the revised manuscript.

Methods:

- Study variables: Please remove the variable number and the data set number.

Authors’ response: Thank you for your suggestion. We have corrected it in the revised manuscript.

Results:

- weren't married: Were not married

- hadn't any information: had no information

- Please remove dot before . (Table 1).

Table 1. Please use one number after coma (18.93%, make it 18.9%). Please revise all table.

Table 1. What does non married mean?

Table 2. Please two numbers after coma. For example: 1.156 (1.014-1.318). It should be 1.15 (1.01-1.31). Please revise all table.

Authors’ response: We have corrected the comments as per your suggestion in the result section of the manuscript. “Non married” in the table one was to mean single mam. In case it made confusion, we changed it to “single” in the revised manuscript.

This manuscript need major editing for language and grammar. At its current form, the language of the article is not appropriate for academic publishing.

Authors’ response: Thank you for your constructive comments to make the manuscript more scientific. We tried to solve the grammatical issue of the manuscript in the revised version.

In the revised manuscript, please give page and line numbers (using continuous options). The authors also have to revise their data availability statement and how can be data accessed. I believe the data will have a repository link.

Authors’ response: Thank you for your constructive comments. We have given the continuous line number in the revised manuscript. The data availability statement is also revised.

---

## [Decision Letter · Decision Letter 2]

15 Nov 2024

Awareness and its determinant factors towards breast examination to detect breast cancer among reproductive age women in Kenya: Multi level analysis of the recent Demographic and Health Survey data

PONE-D-24-24195R2

Dear Dr. Wassie,

We’re pleased to inform you that your manuscript has been judged scientifically suitable for publication and will be formally accepted for publication once it meets all outstanding technical requirements.

Kind regards,

Yitagesu Habtu Aweke, Ph.D

Academic Editor

PLOS ONE

Additional Editor Comments (optional):

Reviewers' comments:

Reviewer's Responses to Questions

**Comments to the Author**

1. If the authors have adequately addressed your comments raised in a previous round of review and you feel that this manuscript is now acceptable for publication, you may indicate that here to bypass the “Comments to the Author” section, enter your conflict of interest statement in the “Confidential to Editor” section, and submit your "Accept" recommendation.

Reviewer #2: All comments have been addressed

2. Is the manuscript technically sound, and do the data support the conclusions?

Reviewer #2: Yes

3. Has the statistical analysis been performed appropriately and rigorously? 

Reviewer #2: Yes

4. Have the authors made all data underlying the findings in their manuscript fully available?

Reviewer #2: Yes

5. Is the manuscript presented in an intelligible fashion and written in standard English?

Reviewer #2: Yes

6. Review Comments to the Author

Reviewer #2: Please revise lines: It was coded as "Yes = 1" if the study participants had awareness of breast examination to detect breast cancer and “No = 0” if otherwise.

7. PLOS authors have the option to publish the peer review history of their article (what does this mean?). If published, this will include your full peer review and any attached files.

Reviewer #2: **Yes: **Muhammad Haroon Stanikzai

---

## [Editor Report · Acceptance letter]

26 Nov 2024

PONE-D-24-24195R2 

PLOS ONE

Dear Dr. Wassie, 

I'm pleased to inform you that your manuscript has been deemed suitable for publication in PLOS ONE. Congratulations! Your manuscript is now being handed over to our production team.

Kind regards, 

on behalf of

PhD Candidate Yitagesu Habtu Aweke 

Academic Editor

PLOS ONE